# Immune Dysfunctions and Immunotherapy in Colorectal Cancer: The Role of Dendritic Cells

**DOI:** 10.3390/cancers11101491

**Published:** 2019-10-03

**Authors:** Sandra Gessani, Filippo Belardelli

**Affiliations:** 1Center for Gender-Specific Medicine, Istituto Superiore di Sanità, 00161 Rome, Italy; 2Institute of Translational Pharmacology, CNR, 00131 Rome, Italy; filippo.belardelli@ift.cnr.it

**Keywords:** colorectal cancer, dendritic cells, immunotherapy, pathogenesis, risk factors

## Abstract

Colorectal cancer (CRC), a multi-step malignancy showing increasing incidence in today’s societies, represents an important worldwide health issue. Exogenous factors, such as lifestyle, diet, nutrition, environment and microbiota, contribute to CRC pathogenesis, also influencing non neoplastic cells, including immune cells. Several immune dysfunctions were described in CRC patients at different disease stages. Many studies underline the role of microbiota, obesity-related inflammation, diet and host reactive cells, including dendritic cells (DC), in CRC pathogenesis. Here, we focused on DC, the main cells linking innate and adaptive anti-cancer immunity. Variations in the number and phenotype of circulating and tumor-infiltrating DC have been found in CRC patients and correlated with disease stages and progression. A critical review of DC-based clinical studies and of recent advances in cancer immunotherapy leads to consider new strategies for combining DC vaccination strategies with check-point inhibitors, thus opening perspectives for a more effective management of this neoplastic disease.

## 1. Introduction

Colorectal cancer (CRC) is one of the major leading cause of cancer-associated mortality worldwide, thus representing an important public health issue, with a great impact in terms of human suffering and costs for the clinical management of patients [1]. The rate of CRC incidence is particularly high in populations living a Western lifestyle, but it is currently increasing in other geographic areas, including low income countries, thus representing a global health challenge [2].

The pathogenesis of CRC exhibits a great level of complexity, being characterized by several multi-step disease events, associated with the accumulation of both genetic and epigenetic alterations of the genome. In fact, the development of CRC is a long process taking several years to progress from barely detectable small neoplastic foci to adenomas and subsequently to malignant carcinomas endowed with metastatic behavior [3].

CRC is characterized by a high heterogeneity given the remarkable genomic instability [4]. Moreover, there is evidence that exogenous factors, such as lifestyle, diet, nutrition, environment and microbiota, contribute to the pathogenesis of CRC, also influencing non neoplastic cells, including immune cells, and leading to further heterogeneity [5,6].

Host immune dysfunctions are important factors contributing to CRC development. Indeed, a significant impairment of the host anti-tumor immunity has been reported during initiation of CRC mostly relying on escape mechanisms adopted by transformed cells to create a favorable growth environment [7,8,9]. During the initial stages of neoplastic transformation and progression, several changes occur within the tumor microenvironment to initially promote neoplastic cell proliferation, subsequently leading either to tumor progression and metastasis or to an immune-mediated cancer inhibition. In particular, the tumor microenvironment can dictate the recruitment of inflammatory and immune cells playing complex roles in either controlling tumor growth or inducing a chronic inflammation status, thus promoting CRC progression by induction of immune suppressive mechanisms [10,11].

Today, an ensemble of data support the statement that inflammation plays an important role in CRC pathogenesis and progression [10]. The low-grade chronic inflammation characterizing obesity, a major risk factor for CRC development, and the anti-inflammatory drug benefits in lowering CRC risk and retarding intestinal tumors in ulcerative colitis patients provide compelling evidence for a link between inflammation and cancer [12]. In this regard, diet is nowadays recognized to play a key role in CRC initiation and progression due to its potential to contribute to a chronic inflammatory condition, either locally in the adipose tissue (AT) or systemically by regulating a variety of immune and inflammatory pathways. In addition, diet strongly controls the composition of the intestinal microbiota that not only maintains the immune homeostasis but can also be involved in colorectal carcinogenesis [13,14,15].

Information stemming from both mouse models and studies in patients points to a key role of immune cells and soluble factors with immunosuppressive activity in the CRC disease process [11]. Among the many cells of the immune system exerting important functions in the host response to neoplastic transformation, dendritic cells (DC) deserve a special attention, since these cells, which are an highly heterogeneous cell population present in the blood, in the lymphoid organs as well as in the tumor microenvironment, represent the major actors in linking innate and adaptive anti-cancer immunity [16].

The clinical management of CRC is firstly based on surgical resection, but the optimal treatments in patients with advanced metastatic disease is still matter of debate. Different protocols of chemotherapy and immunotherapy, including combination therapies, have been used in metastatic patients with variable success [17]. Of note, CRC is one of the first human cancer where a stringent correlation was found between tumor infiltrating CD8^+^ T cells and clinical outcome [18], thus supporting the rationale for evaluating the efficacy of immunotherapy protocols in this neoplastic disease. In fact, the clinical research for implementing the management of CRC patients in an advanced disease stage often included the use of cytokines (i.e., IFN, IL-2), adoptive cell therapy and DC-based vaccines, but variable and inconclusive results were obtained so far. Today, we are facing a momentum of enthusiasm on cancer immunotherapy in the light of the emerging great clinical impact of check-point inhibitors (CPI) [19]. However, major research challenges are to fully understand the mechanisms of the response and to obtain clinical efficacy in non-responding and poorly responding patients by designing more sounded combination therapies. While the role of immunosurveillance in the control of CRC growth and progression is assumed to be of great importance [17,18], patients appear to be resistant to the blockade of immunological checkpoints with monoclonal antibodies (mAbs) specific for cytotoxic T lymphocyte-associated protein 4 (CTLA4), programmed cell death 1 (PDCD1, best known as PD-1) and the PD-1 ligand CD274 (best known as PD-L1), with the exception of a minority of subjects characterized by microsatellite instability (MSI) lesions [20,21]. This has been considered as an apparent paradox and an intriguing issue demanding further research efforts for fully understanding the mechanisms of the resistance to CPI and developing new and more effective therapeutic strategies [22]. In this review, we intend to specifically address the role of DC in the pathogenesis and progression of CRC as well as in the response to immunotherapy. A special attention will be given to the role of microbiota, obesity-related inflammation, diet and host reactive cells, including DC, in CRC pathogenesis, then discussing how we can translate the research progress in this field in strategies of prevention and management of CRC. Likewise, we will review the ensemble of studies reporting the variations of different DC subsets in CRC patients and their correlation with disease stages and clinical outcome. Lastly, we will provide a brief critical overview of the results of DC-based clinical trials in CRC patients, discussing new perspectives for their combination with CPI and some current research challenges for the management of this neoplastic disease.

## 2. Diet, Inflammation and Microbiota in the Pathogenesis of CRC

### 2.1. Diet and Obesity As Important Factors in the Pathogenesis of CRC

Excess adiposity is associated with increased incidence of several cancers and represents an important indicator of survival, prognosis, recurrence and response to therapy in CRC. Notably, patterns and trends in CRC incidence and mortality correlate with geographical location, societal and economic changes and their increase may reflect the obesity epidemic and the adoption of more Western lifestyles. Both genetic and a range of environmental, largely modifiable, lifestyle factors play an important role in CRC etiology. Among these, the links between body weight, dietary patterns and CRC risk are some of the strongest for any type of cancer with profound implications for prevention strategies. It is now well-recognized that CRC risk is highly modifiable through lifestyle, particularly diet and physical activity; recent reports suggest that up to 47% of CRC cases could be prevented by staying physically active, maintaining a healthy body weight and eating a healthy diet (available at https://www.wcrf.org continuous-update-project 2017). Obese subjects have a 1.5–3.5-fold increased risk of developing CRC compared with lean individuals, and epidemiologic evidence indicates that abdominal rather than overall obesity may be more predictive of CRC risk [23]. Multiple changes arising in condition of chronic positive energy balance are likely to contribute to the increased CRC risk and worse outcomes in obesity. In particular, during the progression to obesity, the AT undergoes profound structural and functional modifications [24] tightly coupled with dramatic changes in the immune cell repertoire and functions [25,26], that shift the balance of cell subsets and soluble mediators toward a pro-inflammatory profile. Growing evidence indicates that the chronic low-grade inflammatory state characterizing obesity contributes to the impairment of immune functions, thus representing a key determinant in the development of obesity-related morbidities including cancer [27]. Furthermore, lipids, especially fatty acids (FA), the main components of AT, are recognized to play an important role not only in obesity development but also in the interplay between excessive adiposity and development of associated diseases [28]. In this regard, qualitative changes, rather than quantitative, in the FA composition of AT have been reported to influence tissue dysfunctions and are associated with an enhanced STAT3 activation and concomitant down-regulation of anti-inflammatory pathways such as PPARγ and its downstream target adiponectin [29].

The metabolic disturbances characterizing obesity lead to chronic immune activation as unraveled by the presence of elevated levels of plasmatic inflammatory markers in obese subjects [30]. The bulk of immune alterations observed in obesity may provide an explanation for the higher rate of vaccine failure and infectious disease [31]. In this regard, the white AT, particularly visceral fat, is now well-recognized as a complex immunocompetent organ, homing adipocytes and resident immune cells exhibiting secretory as well as immunological, metabolic and endocrine regulatory activities. Furthermore, AT is a medium- to long-term indicator of FA dietary intake. Among the different factors potentially influencing visceral AT microenvironment and immune cell distribution, the relative composition of ω3/ω6 polyunsaturated fatty acid (PUFA) might play a pivotal role, since these molecules are capable to markedly modulate inflammation and to influence immune functions [32,33]. In this regard, visceral fat adipocytes from obese and CRC subjects exhibit distinct secretory and ω6 PUFA profiles characterized by a prevalence of pro-inflammatory factors and inflammation-promoting FA [34]. Of note, we recently reported that obese and CRC subjects share inadequate dietary habits and altered lipid metabolism, suggesting that the quality of the diet consumed, regardless the quantity of energy intake, is an important aspect to preserve human health [35]. CRC and obese subjects were found to be more prone to follow a saturated fatty acid (SFA)-rich diet and exhibit a reduced content of monounsaturated fatty acids (MUFA) (especially in oleic acid). The composition of AT, in particular in FA, may thus represent an important determinant in shaping the immune cell phenotype and in influencing processes/events occurring in distal tissues that may set the basis for CRC carcinogenesis.

### 2.2. Relationships Between Diet, Microbiota and Immune Dysfunctions in CRC Pathogenesis

The key role of diet in CRC initiation and progression as well as in prevention is not only linked to the capacity of some nutritional components to contribute to a chronic inflammatory condition by regulating a variety of immune and inflammatory pathways, but also to strongly control the composition of the intestinal microbiota. The human microbiota, a collection of commensal microorganisms colonizing gastro-intestinal, genitourinary, oral, respiratory and cutaneous tracts, interacts with the host in different ways and contribute to many important processes such as nutrient absorption, metabolism, tissue development, immunity and tumorigenesis [13,14,15]. It is now well-known that microbiota is influenced by several factors of genetic, dietary and environmental nature. Likewise, some of the metabolic effects of diet rely on gut microbiota. Examples of how diet can influence microbiota emerged from studies in populations consuming different diets, leading to the conclusion that dietary patterns defined as “healthy” (e.g. Mediterranean diet) are associated with higher microbial richness [36]. Among the best associations between diet and gut microbiota are dietary fibers, polyphenols and fats. In particular, different dietary fats may exert different effects on gut microbiota (diversity, alterations of specific microorganisms and their functions) with metabolic consequences such as regulation of systemic low-grade inflammation [37].

In obesity, the gut microbiota displays distinctive features and most studies have demonstrated a reduction in diversity and richness—termed dysbiosis—which has been associated with low-grade inflammation, increased body weight and fat mass, as well as type 2 diabetes (T2D). Nevertheless, the exact microbial signature of a healthy or an obese gut microbiota is still matter of debate. Dysbiosis is associated with a large array of diseases including cancer, where it is implicated in different ways [6]. In addition, microbiota can be directly oncogenic by favoring local mucosal inflammation or systemic metabolic/immune dysregulation or can act indirectly by virtue of its capacity to modulate anti-tumor immunity or the efficacy of anti-cancer therapy. In this regard it is of interest that the abundance of *Akkermansia muciniphila* has been positively associated with the antitumor effect of PD-1 blockade in epithelial tumors [38] and hepatocellular carcinoma [39]. Interestingly, decreased amounts of this bacterium have been linked to obesity, insulin resistance, T2D and other cardiometabolic disorders in rodents and in humans [40]. In addition, alterations of fecal and mucosal microbiota with reduction of bacterial diversity have been reported in CRC patients at different cancer stages [41,42].

In the following chapters, we provide a brief overview of the role of DC in the regulation of inflammatory and immune responses, of their functional changes in CRC patients and of their exploitation in immunotherapy protocols against CRC.

## 3. The Role of DC in the Regulation of the Inflammatory and Immune Responses in CRC

DC represent a heterogeneous group of innate immune cells endowed with the unique capacity to initiate and coordinate the immune response. They are professional antigen presenting cells (APC) and comprise a variety of subsets, of both myeloid and lymphoid origin, as either resident or migrating cells, in lymphoid and non-lymphoid organs. They are able to recognize, capture and process antigens and to present them to naïve T lymphocytes. DC are nowadays recognized as a family comprising several subtypes that differ in ontogeny, gene expression profile, anatomical location, phenotypic and functional features [43]. In this regard, consensus has been recently achieved on the recognition of five major DC types: plasmacytoid DC (pDC), type 1 conventional DC (cDC1), type 2 cDC (cDC2), Langerhans cells and monocyte-derived DC (MoDC). In the steady state, DC are largely present as immature cells exhibiting a high capacity to capture antigen, and a low expression of co-stimulatory molecules and secretion of effector cytokines. The exposure to different stimuli including microorganisms or damaged cells/tissues promotes DC activation, a functional state characterized by a decreased capacity to capture antigen, enhanced expression of MHC class I and II antigens as well as costimulatory molecules, active production of effector cytokines and migration to lymph nodes, where they interact with naïve CD4^+^ and CD8^+^ T lymphocytes.

It is currently thought that DC play an important role in presenting tumor antigens to T cells and in shaping an antitumor immune response, which may result in an effective control of tumor growth [43,44]. However, many studies have revealed how the phenotype and functions of these cells can markedly be affected by several molecular and cellular actors playing complex and even opposite roles within the tumor microenvironment. As an example, there is plenty of evidence indicating that tumors can not only suppress DC maturation, but can also induce the generation of DC endowed with immunosuppressive activities [45,46].

Dietary habits and excessive adiposity can not only influence cancer growth but also shape host immune response [47]. Myeloid DC, but not pDC, have been described to accumulate in the subcutaneous AT of obese subjects. While the number of CD11c^+^/CD141^+^DC is the same in lean with respect to obese subjects, the number of CD11c^+^/CD1c^+^ cells positively correlates with the body mass index (BMI). This accumulation parallels an enhanced presence of Th17 lymphocytes in AT, suggesting a role of DC infiltrating AT in the regulation of tissue inflammation and Th17 cell expansion [31]. Of note, studies carried out in mouse models of obesity suggest that the presence of CD131^+^ DC in the AT of lean mice can be important for the local expansion of T regulatory cells providing anti-inflammatory signals to maintain AT homeostasis [31]. Interestingly, the exposure of immune cells to visceral adipocyte conditioned media from obese and CRC affected subjects favors IL-10 production, reduces the immunostimulatory activity of DC and hampers their capacity to generate γδ T cell-mediated responses induced ex vivo, further highlighting the existence of a regulatory/suppressive AT microenvironment in both obesity and CRC [34]. Furthermore, distinct alterations of the immune cell repertoire in the periphery with respect to the AT uniquely characterize or are shared by obesity and CRC [48].

## 4. Changes in the Phenotype and Function of DC in CRC Patients

Several groups have described qualitative as well as quantitative changes of DC in the blood as well as in the tumor microenvironment of CRC patients at different stages of disease and their possible correlation with the clinical response of patients [46]. The interpretation of the overall results is not always easy, since contradictory data were reported in some cases, possibly due to differences in the clinical settings as well and in the methodologies used to identify specific DC subsets. Here, we will restrict our review to the discussion of only some studies, selected in view of their special potential clinical relevance. The possible correlations between the presence and maturation phenotype of tumor-infiltrating DC with the patient prognosis and clinical response have been investigated by Gulubova and colleagues [49]. These authors found that the presence of CD83^+^ mature DC was lower in the tumor stroma of patients in an advanced disease stage. In general, we can state that negative correlations between the detection of these tumor infiltrating DC and the number of lymph node metastases as well as the survival time of CRC patients were frequently documented [49,50,51]. Notably, by comparing human primary CRC specimens with respect to normal colon mucosa, Schwaab and co-workers found that the number of infiltrating mature DC was higher in the CRC samples, while the DC density in metastases was markedly lower than in CRC primary tumors [52]. Of interest, Michielsen and colleagues reported that tumor conditioned-media from cultured human CRC tissue can impair DC maturation process, possibly by releasing chemokines and other soluble factors capable of inhibiting IL-12p70 secretion by DC [53,54]. Of note, Bauer and colleagues [55] reported that infiltration with mature DC was more elevated in MSI-high (MSI-H) tumors as compared to microsatellite-stable CRC. This observation is interesting since it can provide an explanation for the preferential clinical response of MSI-H CRC patients to novel immunotherapies, including CPI [20,21]. Some groups have also investigated the number and phenotype of DC in the peripheral blood of CRC patients with respect to healthy individuals as well to the disease stage and progression [56,57,58,59]. In particular, it has been found that the number and functions of different blood DC subsets were impaired in CRC patients, demonstrating that the magnitude of these effects positively correlated with the disease stage and prognosis [59]. Similar results were obtained by Orsini et al [57], who described a significant reduction of the DC number in total and advanced stage-CRC patients compared to healthy controls, and reported that this reduction was totally recovered after complete tumor resection, further supporting the concept of the importance of systemic immunosuppressive effects exerted by the tumor toward circulating blood immune cells. Of interest, some authors have also reported that the reduction in DC was mostly due to changes in pDC population [57].

A useful in vitro model to investigate the biology of DC and the mediators and mechanisms important in shaping their functions is represented by MoDC, generated from monocytes by in vitro treatments with GM-CSF and various cytokines, such as IL-4, IFN and other activation/maturation factors. Thus, some published studies where the phenotype and functions of MoDC from CRC patients were compared to those detectable in control subjects are available [56,60,61,62]. In particular, Orsini and colleagues showed an impaired in vitro differentiation of CRC patients’ monocytes into immature DC, compared to healthy subjects [57]. Of note, CRC MoDC exhibited a reduced expression of costimulatory molecules and an impaired ability to present antigens to allogenic T lymphocytes and to stimulate proliferation, together with an immunosuppressive cytokine profile, mostly characterized by increased IL-10 and reduced IL-12 secretion [57]. Of interest, it was reported that the maturation status of the MoDC from CRC patients was phenotypically and functionally superior to the in vivo blood DC recovered from the same individuals. This observation somehow supported the potential value of using MoDC from CRC patients for clinical studies of cancer immunotherapy [63].

## 5. DC and Immunotherapy of CRC

Since the early study by William Coley in cancer patients treated with killed bacterial vaccine in 1891, for more than 120 years, the history of cancer immunotherapy has been characterized by alternate cycles of optimism and discouragement. The clinical use of certain cytokines (i.e., IFN-α and IL-2), the subsequent identification of the first set of human tumor antigens, the progress on cancer vaccines and in the development of protocols based on adoptive cell therapy have all represented important milestones in the field of cancer immunotherapy. However, it is only in recent years that we have registered a fundamental progress, which today leads to consider cancer immunotherapy as the latest revolution in cancer therapy. This is mainly due to the impressive results achieved in patients with different type of malignancies after treatment with CPI [19]. With regard to CRC, however, only modest clinical effects have been observed so far in patients treated with these new immunotherapy drugs (including anti-CTLA-4, anti-PD1 and anti-PD-L1 antibodies) [64], which instead proved to be highly effective in other human malignancies (including melanoma, Hodgkin lymphoma and non-small lung cell cancer).

In view of their crucial role in linking innate and adaptive antitumor immunity, DC have extensively been used in cancer immunotherapy clinical trials over the last two decades [44,65]. Notably, the large majority of DC-based studies involved the use of patient-derived DC generated from peripheral blood monocytes differentiated in vitro by the addition of cytokines (generally GM-CSF and IL-4), loaded with tumor-derived antigens by different experimental procedures and subjected to a further step of in vitro maturation, before their injection in therapeutic vaccination protocols [65]. In 2011, the registration of the DC-based Provenge vaccine for patients with prostate cancer led to a transient momentum of special optimism for the clinical development of DC-based cancer vaccines. However, in the following years, this cancer vaccine was not further developed and, in view of the limited response observed in hundreds of clinical trials, the clinical development of DC-based vaccines was regarded with a lower attention with respect to that devoted to new emerging tools in cancer immunotherapy, such as CPI and CAR-T adoptive cell transfer. There are recent and comprehensive reviews reporting the results of DC-based clinical trials, which also critically discuss the major challenges for their clinical development [44,65]. While the lack of any relevant toxicity represents a good starting point, there are still several critical issues to be addressed, including identification of the optimal DC to be used, reliable criteria to characterize the quality and potency of these cell products, the source/loading of tumor antigens, the modalities of injection and the possible combinations with other drugs/treatments to increase their clinical efficacy. Today, we are facing a renovated interest in the development of new generation DC-based vaccines, as a result of a better understanding of the DC biology and of the discovery of new immunomodulatory molecules expected to enforce cancer immunotherapies [66,67,68].

In considering new and potentially more effective DC types to be used in cancer immunotherapy protocols, we may consider to use DC generated by monocytes by a short-term in vitro exposure to IFN-α and GM-CSF [69]. In fact, these DC (named as IFN-DC) exhibit a unique attitude to take-up tumor apoptotic bodies and induce a potent tumor specific T cell immunity in preclinical models [70] as well as in cancer patients, as suggested by results in pilot clinical trials where IFN-DC have been inoculated intratumorally in patients with metastatic melanoma [71] and indolent lymphomas [72].

Table 1 reports the main published data of clinical trials based on the use of DC in CRC patients. The general messages stemming from an overview of the main results published so far can be summarized as follows: (i) the large majority of studies reported results of pilot phase I-II trials in metastatic CRC patients with a relatively small number of patients; (ii) different methodologies were used for the in vitro generation of DC-based vaccines from monocytes, including the use of various cytokines and other activation/maturation factors, rendering difficult the comparison of the results; (iii) different methods of tumor antigen loading of DC were utilized and, in a few cases, unloaded DC were used; (iv) the regimen and route of DC administration as well as the number of DC injected markedly differed among the published studies; (v) in some cases, the patients were also treated with either conventional (for instance chemotherapy) or additional experimental cell therapies; (vi) there were marked differences in the protocol design as well as in the immunomonitoring methods to evaluate DC-induced immunogenicity. All this suggests that, even though some of these trials have represented important proof-of-principles for the lack of toxicity and potential efficacy of DC-based vaccines in inducing antitumor immune responses in CRC, the translation of the possible use of DC for the development of new-generation strategies of CRC immunotherapy needs further and coordinated research efforts.

One of the major reason for the limited response of CRC to the immunotherapy is thought to be represented by the immunosuppressive tumor microenvironment which generally occurs in patients with advanced disease. As a matter of fact, the major challenge for developing an effective protocol of cancer immunotherapy is indeed to counteract the several and complex immunosuppressive mechanisms activated in the tumor microenvironment of cancer patients. The role of several cancer-induced immunosuppressive mediators in CRC prognosis and treatment response has been reviewed elsewhere [9,10,11]. These mediators include cells endowed with immunosuppressive activity, such as regulatory T cells and certain macrophage populations, as well as soluble factors. Notably, modulations of the local production of certain cytokines as well as in their response can play a role in shaping the type of antitumor response [85]. Of interest, loss of type I IFN receptor has recently been identified as an important key factor linked to tumor microenvironment immunosuppression in CRC patients [86]. Thus, we may assume that a local production of and response to cytokines such as type I IFN can exhibit a beneficial role in shaping the response towards an effective immune control of CRC.

Today, in the new era of CPI, major research challenges are to fully understand the mechanisms of the response and how to increase the clinical efficacy in poorly responding patients by designing more sounded combination therapies, which may also include DC. Of interest, a recent study showed that an effective antitumor response to anti-PD1 mAbs strictly requires the occurrence of intratumoral DC [68]. Likewise, some recent studies have added further evidence underscoring a previously underestimated role of intratumoral DC in the tumor microenvironment in mediating the clinical response to immunotherapy regimens in cancer patients [87,88]. Of interest, the intratumoral DC involved in the generation of an antitumor response to anti-PD1 mAbs were characterized as mature DC producing high levels of IL-12 [68]. Notably, IFN-DC, which undergo a rapid and complete maturation after peripheral blood lymphocyte co-cultivation, are high producers of IL-12 [89] and therefore may represent good candidates for potentiating anti-PD1-based therapies. Of interest, we had previously shown that IFN-DC are highly efficient APC in inducing both CD8^+^ and CD4^+^ T-cell-mediated responses against the colon tumor antigen-1 in CRC patients at different stages of the disease [90]. Thus, on the basis of the overall preclinical and clinical data on IFN-DC obtained by our group [69,70,71,72,89,90], we consider these DC as valuable autologous cell products for the development of new-generation DC products to be used in clinical trials in CRC patients. For these DC-based therapies, we may envisage therapeutic scenarios where CRC patients are treated with autologous DC, either as unloaded APC injected intratumorally (endogenous tumor vaccination) in combination with agents either inducing or enhancing tumor cell death [71,72], or as in vitro antigen loaded DC, and subsequently injected with anti-PD1 antibodies or other CPI to increase the antitumor response in selected combination therapies.

## 6. Conclusions

CRC represents one of the human malignancies where promotion of prevention strategies can play a major role in reducing cancer development and tumor burden and progression. In fact, primary prevention, based on special attention to reduce exposure to environmental and lifestyle risk factors (including diet and physical exercise) is indeed of great importance for reducing CRC incidence, with enormous impact in terms of public health and reduction of costs for the national health services. In addition, in view of the long and multi-step processes involved in CRC development, strategies of secondary prevention, including the promotion of the use of early diagnostic platforms, can be very important for prevention and control of CRC. In spite of all this, the optimal therapeutic management of patients with metastatic CRC remains an important issue in clinical oncology. While surgery and chemo-radiotherapy interventions continue to represent essential therapeutic options depending on the stage of the disease and the clinical settings, immunotherapy has recently emerged as a powerful tool for tertiary immune prevention.

Recently, we have learned new rationales and modalities for combining different immunotherapy regimens with both conventional and target therapies. Likewise, we have recently started to understand the importance of sex- and gender-specific differences in several pathologies including cancer. In fact, gender disparities have been reported in different aspects that can collectively influence CRC pathogenesis and therapy [91]. Thus, CRC incidence, outcome and survival as well as microbiota composition exhibit a different trend in men and women [92]. Likewise, some of the main risk factors for CRC, such as obesity and lifestyle-related aspects (i.e. diet and physical activity), are strongly linked to gender [93,94]. Worth of note, differences in the immune response have also been reported in women and men [95]. However, at the moment there are no studies describing gender differences in DC functions in CRC patients. Studies on gender related immune dysfunctions in CRC taking into consideration the DC biology are expected to contribute to our understanding of the pathogenesis and to the clinical management of this neoplastic disease.

Figure 1 summarizes the main DC dysfunctions observed in a specific non-neoplastic tissue relevant for CRC pathogenesis (i.e., AT), blood and tumor microenvironment of CRC patients, along with the risk factors playing a role in the disease process. It also depicts some main strategies and challenges for the development of DC-based immunotherapy strategies in CRC patients. Such strategies are aimed at considering DC either as in vivo targets for tumor antigen delivery and/or for recruiting and activating DC within the tumor microenvironment, or as autologous cell products generated from monocytes by different in vitro manipulations and then reinfused into the patients. In any case, new generation immunotherapy strategies should consider what is the impact and possible role of DC, which represent important cell actors in CRC pathogenesis and antitumor immune-based control.

How to reverse DC dysfunctions occurring at different disease stages and in various tissues in CRC patients still remains a complex issue deserving further research efforts. In principle, intervention strategies for restoring DC functions in the very early stages of the disease could also be considered, but we need to reach a better knowledge of the role of this highly heterogeneous cell population in the pathogenesis and progression of CRC. The recent advances on the cross-talk between gut microbiota and human health and on the potential of lifestyle, food components and/or dietary patterns to modulate this functional interplay has opened new perspectives for diet-based interventions in the modulation of the antitumor immune response [96].

With regard to the still critical issue of implementing the management of advanced metastatic CRC, a great importance is currently given to combination therapies, since we now have a much better knowledge on how different therapeutic tools and strategies should be associated. With the advent of next-generation sequencing methodologies, we have now the unprecedented ability to identify tumor, host, and microbial genomes. The growing application of these novel technologies to finely characterize patient’s tumor and driver mutations as well as the immune repertoire for evaluating genetic responses to current immunotherapies has opened new ways to maximize patient benefits through cancer precision medicine strategies. We conclude by stating that, taking into account some recent findings [68,87,88], new generation DC-based strategies can represent a promising added value for enhancing the response to the new therapeutic regimens, including CPI, in CRC patients.

## Figures and Tables

**Figure 1 cancers-11-01491-f001:**
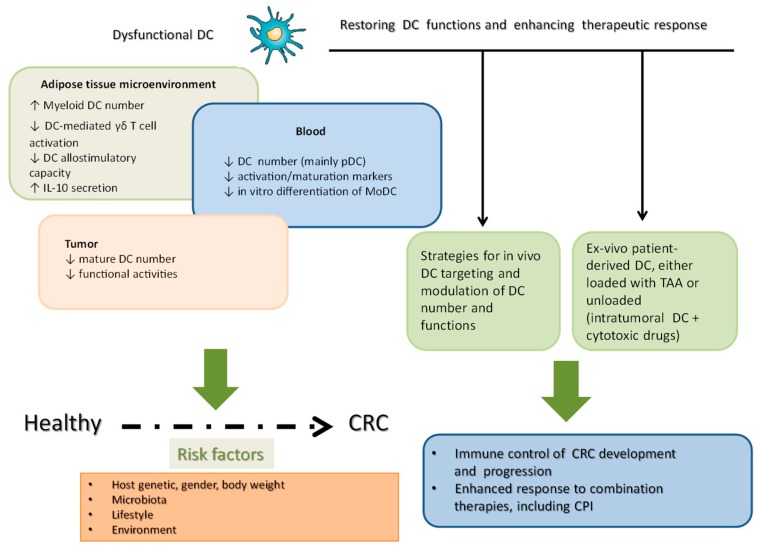
A schematic representation of the main DC dysfunction in adipose tissue, peripheral blood and tumor tissue highlighting the main strategies to restore DC functions and to enhance anticancer immune response.

**Table 1 cancers-11-01491-t001:** Main clinical studies aimed at evaluating DC-based therapies in CRC patients.

Patients	DC generation	Ag loading	DC administration	N	Major findings	Ref.
Metastatic, CEA^+^, HLA-*0201; Phase I	GM-CSF/IL-4, + TNFα, PGE2, IL-1β	CEA altered peptide	1–5 × 10^7^, i.v.; 4 times, every 2nd week	7	In vivo expansion of peptide-specific CD8^+^ T cells	[73]
Metastatic, CEA^+^; Phase I	GM-CSF/IL-4	Fowl-pox vector encoding rCEA and costimulatory molecules	5 × 10^5^; s.c./i.d; 1 or 2 cycles of 4 weekly injections	11	Induction of CEA-specific T cells; trend of correlation with clinical response	[74]
Metastatic, HLA-A2^+^, Phase I	IL-13/GM-CSF, maturation factors	6 CEA peptides	35 × 10^6^, i.d., 4 injections every 3 weeks	11	Progressive disease in spite of T cell response to tumor associated antigens	[75]
Metastatic, after resection of metastases; Phase I-II	GM-CSF/IL-4	Autologous tumor lysate, KLH	5 × 10^6^ into 2 inguinal lymph nodes under ultrasound guidance; week 1, 3 and 6	26	Tumor specific T cell response (63%); correlation with recurrence-free survival; no difference if DC were further treated or not with CD40L	[76]
Metastatic, CEA^+^, HLA-A*2402; Phase I-II	IL-4/GM-CSF/IFNα, streptococcus pyrogenes	CEA peptide	11–115 × 10^6^, s.c., 2-8 injections	8	Trend of correlation between CEA-specific cytotoxic T cells and clinical efficacy	[77]
Metastatic, after metastasis resection; Phase II	IL-4/GM-CSF	Poxvectors encoding CEA, MUC-1 and costimulatory molecules	10^7^, s.c./i.d. 3 times per month/3 months; comparison with patients injected with poxvectors + GM-CSF	37	Both DC-poxCEA and poxCEA +GM-CSF treatments showed similar response; longer survival time compared to contemporary unvaccinated group	[78]
Stage Dukes B2 and Dukes C; Phase I-II	IL-4/GM-CSF	TCL, rCEA protein	5 × 10^6^–2 × 10^7^, s.c.; days 1, 14, 28, 56	12 ^	Suggestion of clinical effect with TCL-DC, but no effect with CEA-DC	[79]
Metastatic, after resection of metastases; pretreatment with low dose chemotherapy; Phase I-II	IL-4/GM-CSF	TCL	Average DC dosage: 188 × 10^6^, s.c.; 3–5 injections in 2 weeks; patients also received i.v. injections of CIK cells	13	Reduction of post-operative disease risk; increase of overall survival	[80]
Metastatic, unresectable; Phase II	IL-4/GM-CSF/TNFα	TCL	10^7^, i.v., for the first 3 weeks; i.d. for the last 3 weeks; i.v. CIK cell infusions for 4 days	100	DC/CIK therapy can induce anti-CRC immune response (DTH) with a potential impact on survival and quality life with respect to control group	[81]
Metastatic, resistant to standard therapies; Phase I-II	IL-4/GM-CSF, + maturation factors	rCEA protein	10^6^, s.c., mixed with tetanus toxoid; 3 other s.c. injections of the same DC number	12	T cell reactive against CEA in 2 patients; 2 patients with stable disease; 10 patients showed progression; need to enhance antitumor T cell response	[82]
Metastatic, phase II; DC vaccine + best supportive care versus best supportive care	IL-4/GM-CSF + maturation factors	Autologous TCL	5 × 10^6^ (1, 10, 20, 40, 120 days), s.c.	28	Induction of tumor specific T cell response; no increase of overall survival with respect to the “best supportive care” group	[83]
Metastatic, resistant to standard therapies; Phase I-II	GM-CSF + killed BCG mycobacteria + IFNα	No in vitro antigen loading	2–15 × 10^6^; 2–6 injections, i.t. using image guidance	7	Cytokines produced by DC (IL-8 and IL-12p40) correlate with clinical outcome	[84]

Abbreviations: N: Patients’ number; TCL: tumor cell lysate; rCEA: recombinant CEA; s.c.: subcutaneous; i.d.: intradermal; i.v.: intravenous; i.t.: intratumoral; CIK.: cytokine-induced killer cell. ^6 out of 12 patients injected with DC-loaded TCL, 6 with CEA.

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
