# Peer review of "Immune Dysfunctions and Immunotherapy in Colorectal Cancer: The Role of Dendritic Cells"

_cancers, 2019, doi:10.3390/cancers11101491_

Round 1

Reviewer 1 Report

The work is well structured and is complete. I would suggest the authors to investigate the role of the microbiota in stimulating the immune system in more detail. It is known that intestinal dysbiosis could significantly influence the response to immunotherapy. In particular, the important role played by Akkermansia muciniphala in the response to immunotherapy is emerging from numerous studies. Moreover, in obese subjects there is a loss of enrichment of Akkermansia, thus promoting chronic inflammation and, in the long term, the development of colorectal cancer. I conclude by saying that the work is very well done and could be more fascinating if the role of Akkermansia in the modulation of the response to immunotherapy is highlighted.

Author Response

As a follow up to the Reviewer’s comment the role of Akkermansia in the modulation of the response to immunotherapy has been briefly discussed at page 4, lines 167-171, with the inclusion of additional references.

Reviewer 2 Report

This is a very well written review about the role of immune cells (especially dendritric cells) in CRC.

Most aspect of immune regulation of CRC are covered. However, there are already many reviews about this topic.

Perhaps the authors could add a short paragraph reviewing how novel technologies -for example NGS- are generating relevant data that will help understand and devise strategies to tackle the immunological phenomena in CRC.

Author Response

We have taken into account the Reviewer’s suggestion. However, since the topic is rather vast, we have preferred just to quote this aspect in the discussion (page 12, lines 416-421) instead of adding a paragraph.

Reviewer 3 Report

This is a very well-written review article by Gessani et al covering recent progresses in the etiology and treatment of colorectal cancer, and the link between dysfunctional dendritic cells and cancer development/therapy. The authors summerized nicely our current knowledge on diet, inflammation and the pathogenesis of CRC, and linked these environmental and life-style factors to dysfunctional DC. The authors also summarized current clinical trials and their results in DC-mediated cancer therapy, and pointed out future directions for the improvement of CRC immunotherapy. Overall the work was flawless and well done. 

Author Response

We thank the Reviewer for appreciating our work.